# Statistical Distributions of Genome Assemblies Reveal Random Effects in Ancient Viral DNA Reconstructions

**DOI:** 10.3390/v17020195

**Published:** 2025-01-30

**Authors:** Fernando Antoneli, Cristina M. Peter, Marcelo R. S. Briones

**Affiliations:** Center for Medical Bioinformatics, Escola Paulista de Medicina, Federal University of São Paulo (UNIFESP), São Paulo 04039-032, SP, Brazil; fernando.antoneli@unifesp.br (F.A.); cristina.peter@unifesp.br (C.M.P.)

**Keywords:** ancient DNA, genome assembly, ancient viruses, statistical distributions, power laws, log-normal laws

## Abstract

Ancient human viruses have been detected in ancient DNA (aDNA) samples of both Anatomically Modern Humans and Neanderthals. Reconstructing genomes from aDNA using reference mapping presents numerous problems due to the unique nature of ancient samples, their degraded state, smaller read sizes and the limitations of current methodologies. The spurious alignments of reads to reference sequences (mapping) are a main source of false positives in aDNA assemblies and the assessment of signal-to-noise ratios is essential to differentiate bona fide reconstructions from random, noisy assemblies. Here, we analyzed the statistical distributions of viral genome assemblies, ancient and modern, and their respective random “mock” controls used to evaluate the signal-to-noise ratio. We tested if differences between real and random assemblies could be detected from their statistical distributions. Our analysis shows that the coverage distributions of (1) real viral aDNA assemblies of adenovirus (ADV), herpesvirus (HSV) and papillomavirus (HPV) do not follow power laws nor log-normal laws, (2) (ADV) and control aDNA assemblies are well approximated by log-normal laws, (3) negative control parvovirus B19 (real and random) follow a power law with infinite variance and (4) the mapDamage negative control with non-ancient DNA (modern ADV) and the mapDamage positive control (human mtDNA) are well approximated by the negative binomial distribution, consistent with the Lander–Waterman model. Our results show that the tails of the distributions of aDNA and their controls reveal the weight of random effects and can differentiate spurious assemblies, or false positives, from bona fide assemblies.

## 1. Introduction

The field of paleovirology research relies on the detection of viral genomes embedded in DNA and the raw sequencing data of its hosts. Because of smaller genomes and the scarcity of integrated copies, sequence reads of these pathogens tend to be smaller than the average sequence reads of the hosts. Genome remnants of ancient viruses have been detected in ancient DNA (aDNA) samples ranging from the Middle Ages to the Paleolithic [1,2]. Reconstructing aDNA using genome mapping presents numerous challenges due to the unique nature of ancient samples, their degraded state and the limitations of current sequencing methodologies. These artifacts might produce spurious alignments in aDNA genome assemblies with even greater weight than in modern DNA assemblies. Therefore, the field of paleovirology is highly dependent on accurate reconstructions of ancient viral genomes obtained from aDNA data. The genomic ‘fossil record’ helps us understand virus-host interactions over evolutionary timescales [3], enabling the understanding of ancient viral diversity [4] and giving us the ability to decipher the ancient evolution of filoviruses and interactions with vertebrate hosts [5].

Spurious alignments in genome assemblies occur when sequences are incorrectly aligned to the reference genome due to various technical or biological factors [6,7]. These misalignments can lead to errors in genome annotation, variant calling, or downstream analyses. Common causes and contexts for spurious alignments are as follows: (1) repetitive sequences, such as highly repetitive regions (e.g., transposable elements, satellite DNA), can cause reads to align to multiple loci, leading to ambiguous or incorrect placements, (2) paralogous regions, or sequences that are similar due to gene duplication events (paralogs), can align to incorrect paralogous loci instead of their true origin, (3) low-complexity regions such as regions with simple sequence repeats (e.g., homopolymers, di-/tri-nucleotide repeats) often cause misalignments because they lack unique sequence context, (4) errors introduced during sequencing, such as substitutions, insertions, or deletions, that can distort the sequence and lead to incorrect alignments, (5) poor reference quality, such as incomplete or inaccurate reference genomes, can result in reads aligning to incorrect locations or being mapped to scaffold gaps, (6) cross-species contamination, when reads originating from contaminant DNA (e.g., symbionts, pathogens, or laboratory contamination) may spuriously align to the closest matching sequences in the reference genome and (7) inversions, translocations, or structural variants, when large structural rearrangements can mislead mapping algorithms, causing reads from one genomic context to align to a different one [8].

False positives in variant calling can be caused by spurious alignments, when misalignments create the appearance of SNPs, indels, or structural variants that are not truly present in the sample [9]. Also, the misannotation of genes due to the incorrect alignment of reads leads to errors in gene prediction or expression quantification, and assembly gaps and chimeric contigs can be produced by misplaced reads that contribute to assembly errors such as artificial contigs or scaffolds [10]. The minimization of artifacts caused by spurious alignments can be obtained by improvements in mapping algorithms, masking repetitive elements, the filtering of low-quality reads, stringent parameters, alternative reference genomes and post-mapping quality control. In cases of extreme complexity, performing de novo assembly can help to reconstruct genomic regions without reliance on a reference genome and de novo assembly [11].

In the case of ancient DNA, the challenge of genome assembly is even greater. Reconstructing aDNA using genome mapping presents numerous problems due to the unique nature of ancient samples, their degraded state and the limitations of current methodologies [6]. The main problems with aDNA are the following: (1) DNA degradation by fragmentation, often into short pieces (~30–100 base pairs), making it difficult to map accurately to the reference genome, (2) chemical damage such as cytosine deamination causing C-to-T or G-to-A transitions, particularly at fragment ends, introducing errors in alignments and variant calling and (3) low complexity, where some degraded regions lose complexity and are difficult to align uniquely [12].

All mainstream methods of DNA sequencing rely on reading fragments of DNA (reads) that are usually much smaller than the genome to be sequenced and assembled by mapping to a reference. The common abstraction to these methods is that of a mathematical covering problem. In 1988, Lander and Waterman published a study examining the covering problem which is still used as a guideline to estimate the desired sequencing coverage [13]. In the Lander–Waterman model, the basic statistical assumption is that reads are generated uniformly, at random, from the genome, known as the homogeneity assumption. In the homogeneous model, the coverage of each base pair follows a Poisson distribution. This distribution, however, imposes a severe restriction because it excludes the possibility of overdispersed coverage distributions.

When heterogeneity is considered, the coverage of each base pair follows a Poisson mixture with a latent distribution belonging to the gamma distribution family. Then, the number of reads covering a base pair follows a Poisson–gamma distribution, also known as a negative binomial distribution. This is a family of distributions parameterized by two positive real numbers (*r*, *μ*), where *r* is the dispersion parameter and *μ* is the mean value. When *μ*/(*μ* + *r*) tends to 0 and *r* tends to infinity, in such a way that *μ* tends to a fixed limit *μ*0, the negative binomial distribution approximates a Poisson distribution with rate *μ*0. Therefore, the negative binomial distribution is the simplest generalization of the Poisson distribution that allows for overdispersion. Finally, it is important to note that the Poisson distribution and the resulting negative binomial distribution are light-tailed distributions, that is, far from power laws [14]. Based on these considerations, it can be proposed that the problem of quality in genome assemblies by mapping to a reference can be, at least in part, examined from the perspective of the distributions of the reads mapped to the reference. It seems that the parameters of these distributions can be affected by the randomness caused by spurious mapping of reads (or the other problems affecting genome assemblies, as discussed above). The comparison and analysis of distributions, and their properties, might reveal the level of randomness and assess the quality of genome assemblies.

Accordingly, here we analyzed the statistical distributions of viral genome assemblies, ancient and modern, and their respective random “mock” controls as defined previously to evaluate the signal-to-noise in aDNA assemblies [2]. We conclude that the tails of the distributions of aDNA and their controls reveal the weight of random effects in assemblies and can differentiate false positive assemblies, caused by spurious alignments, from bona fide aDNA genome assemblies.

## 2. Material and Methods

### 2.1. Theoretical Background

#### 2.1.1. Distributions with Heavy Tails

The univariate distributions can be divided into two classes: the heavy-tailed and the light-tailed. The heavy-tailed distributions are characterized by the property that their tails decay more slowly than exponentially. The light-tailed distributions are characterized by the property that their tails decay at an exponential rate, or faster. This distinction is important in modeling real world phenomena because a heavy-tailed distribution (such as Cauchy distribution, a power law) has a greater probability of rare events (larger deviations from the mean) than a light-tailed distribution (such as a Gaussian and exponential) (Figure 1) [15,16]. This means that the heavier the tail the larger the random effects.

The exponential distribution, the gamma distribution and the normal (or Gaussian) distribution are examples of light-tailed distributions. The log-normal, the Pareto distribution and the Cauchy distribution are examples of heavy-tailed distributions.

The class of heavy-tailed distributions is quite vast and general which makes it difficult to work with. Therefore, many different narrower and more tractable subclasses of heavy-tailed distributions have been introduced. The two most important such subclasses are the sub-exponential distributions and the regularly varying distributions. The term “fat-tailed” in the literature does not have any rigorous definition. Depending on the research community, the terms fat-tailed and heavy-tailed are synonymous or that the fat-tailed is a subset of heavy-tailed. Here, we consider “fat-tailed” as a synonym of regularly varying [17,18].

The large majority of commonly used heavy-tailed distributions are, in fact, sub-exponential, including the log-normal and the Pareto distributions. However, what distinguishes these two examples is that the Pareto distribution is regularly varying whereas the log-normal is not. The Pareto distribution is an example of a continuous power law probability distribution; that is, it describes a quantity whose probability density decreases as a power of its magnitude. Power laws are the distributions with the heaviest tails and have the important property of scale invariance.

For the practical purpose of determining if a given real-world empirical distribution is a power law, there is an advantage in considering not only the pure power laws, but their “perturbations”, as well. The class of regularly varying distributions is very convenient to work with because it not only contains the “pure power laws”, such as the Pareto distributions, but is much larger. Particularly, it contains all the distributions that deviate from pure power laws by means of a slowly varying function; that is, a function that varies slowly at infinity, with classic examples including functions converging to constants or the powers of logarithmic functions. This definition allows the distribution to deviate from a pure power law arbitrarily but without affecting the power law tail exponent [17].

Complex stochastic processes driving the evolution of many different real-world phenomena can hardly produce perfect power law dependencies without any deviation from a pure power law [19]. Searches for pure power law dependencies in real-world data revealed that this distribution is exceedingly rare [20]. Therefore, it is important to consider the full class of regularly varying distributions instead of the pure power laws.

#### 2.1.2. Power Law Estimation

The proper estimation of the tail exponent, under the assumption that a given empirical distribution is a regularly varying distribution, is a hard problem to solve. This problem has attracted extensive attention in probability, statistics, physics, engineering and finance, where a variety of estimators have been developed for this task, all based on Extreme Value Theory [21].

We adopted the method of Voitalov et al. which consists of 3 estimators: Adjusted Hill (H), Moments (M) and Kernel (K) [22]. These are, currently, the only existing estimators that satisfy the following criteria: (1) applicable to any regularly varying distribution, (2) statistically consistent, i.e., have been proven to converge to the true tail exponent if applied to increasing length sequences sampled from any regularly varying distribution and (3) fully automated by the means of the ‘double bootstrap method’, which has been proven to yield the optimal estimation of the tail exponent for any finite sequence of numbers sampled from any regularly varying distribution.

It is important to stress that based on any given finite sample, there is absolutely no way to tell how likely the hypothesis is that it was sampled from a regularly varying distribution. In view of this impossibility, the best strategy is to simply rely on the estimates of the Adjusted Hill (H), Moments (M) and Kernel (K) estimators.

If the estimator results are all positive, for a given sample, then it might be the case that the empirical distribution comes from a regularly varying distribution. Yet if these estimates are negative or close to zero, then the chances of that are vanishingly small. However, there is no, and cannot be any, rigorous way to quantify these chances, using hypothesis testing or any other methods. In view of these considerations, Voitalov et al. [22] take the conservative approach, and propose the following definition of an empirical power law distribution, based on the values of the 3 estimators above: (1) an empirical distribution is ‘Not Power Law’ (NPL) if at least one estimator returns a negative or zero value, (2) an empirical distribution is ‘Hardly Power Law’ (HPL) if all the estimators return positive values, and if at least one estimator returns a value ≤ 1/4, (3) an empirical distribution is ‘Power Law’ (PL) if all the estimators return values > 1/4, (4) power law distributions having divergent second moments, meaning that the tail exponent is <3, i.e., infinite variance, are of particular interest and (5) a power law empirical distribution has a ‘Divergent Second Moment’ (DSM) if all the estimators return values > 1/2. Finally, it is important to note that there are no restrictions on how close to each other the estimated values must be in the definitions above.

#### 2.1.3. Empirical Coverage Distributions

The empirical coverage distribution is a discrete probability distribution *P*(*k*) defined on the non-negative integers and is obtained from a genome assembly by mapping to a reference sequence by counting, for each *k* = 0, 1, 2, …, how many bases are covered by *k* reads. The expected number of covered bases is the mean of this distribution. It is convenient to consider the log-transformed distribution (LTD), obtained by replacing *k* by (log *k*) (natural logarithm) in the above. This allows one to compare with a (discretized) Normal Distribution using a quantile–quantile (Q-Q) plot. It is also common to consider the log-log representation of distribution given by (log *k*, log *P*(*k*)). Finally, one can define the complementary cumulative distribution function (CCDF) associated with *P*(*k*) by *Ḟ*(*k*) = 1 − *F*(*k*), where *F*(*k*) is the cumulative distribution function (CDF) associated with *P*(*k*).

### 2.2. Genome Assembly Data

Genome data analyzed here are fully characterized elsewhere [2]. BAM files of the assemblies in [2] were used for coverage calculation using Geneious Prime 2024 software (https://www.geneious.com, accessed on 22 March 2024). Coverage data were exported to a csv file and reordered as “number of sites as a function of coverage”.

We considered the four ancient DNA (aDNA) virus assemblies from [2] and their corresponding random “mock” reference assemblies: (a) Neanderthal adenovirus virus reference (ADV) and random reference (ADV-R), (b) Neanderthal herpesvirus reference (HPV) and random reference (HPV-R), (c) Neanderthal papillomavirus reference (HSV) and random reference (HSV-R), and (d) the negative control parvovirus B19 reference (PB19) and its corresponding random reference (PB19-R).

Two mapDamage controls consisting of a human mitochondrial DNA reference (mtDNA) mapped to aDNA reads (positive control) and modern adenovirus (MADV) reads mapped to an ADV reference (negative control) from [2] were also analyzed.

### 2.3. Analysis of Coverage Distributions

For each of the four ancient virus assemblies described in item 2.2., the empirical coverage distribution was computed and analyzed by the program TIE (Figure 1d). Subsequently, we used the program PLFit to estimate, in the cases where a power law was excluded, if the empirical coverage distribution can be modeled as a log-normal distribution. For each of the two non-ancient assemblies described in Section 2.2., the empirical coverage distribution was computed; it was compared using a two sample Kolmogorov–Smirnov test, with a simulated Negative Binomial Distribution, as predicted by the Lander–Waterman model. We used the R package ‘KSgeneral’ version 2.0.2 which allows for the comparison of discrete distributions, i.e., ties (repeated observations) are allowed [23].

### 2.4. Software

The computation of the estimators for the classification of power law distributions is performed by a Python program called ‘Tail Index Estimation’ (TIE) (https://github.com/ivanvoitalov/tail-estimation, accessed on 30 September 2024) [22].

The program ‘PLFit Algorithm’ [20,24] was used as implemented in R package ‘poweRlaw’ [25] to test for the possibility of a non-power law distribution to be well approximated by another heavy-tailed distribution. There are two tests implemented in the R package ‘poweRlaw’: Vuong’s test [24] and a Bootstrap test [25].

## 3. Results

### 3.1. Coverage Distributions

The coverage distributions obtained from BAM files of all assemblies analyzed (ADV, HSV, HPV, PB19 and the respective random controls) are depicted with their corresponding curve fittings (Figure 2). The corresponding histograms and Q-Q plots of the log tran-formed coverage are shown in Figure 3.

The coverage distributions were used to calculate the log-log plots of the distributions (log *k*, log *P*(*k*)) and the log-log plots of the complementary cumulative distribution functions (log *k*, log *Ḟ*(*k*)) (Figure 4).

The plots of the estimators are shown in Appendix A. The Adjusted Hill (H) estimator and its smoothed version are depicted in the original scale and in log-scale for the number of simulation steps (number of order statistics *k*). The Moments (M) and Kernel (K) estimators are depicted in their original scales and in the log-scale for the number of simulation steps (number of order statistics *k*) (Appendix A).

### 3.2. Basic Statistical Parameters of the Coverage Distributions

The basic statistical parameters of the coverage distributions depicted in Figure 1 (mean, median, standard deviation and the number of mapped reads of each assembly) are shown in Table 1. These parameters are used in the Welch test for estimating the signal-to-noise as previously described [2].

The log-transformed distribution (LTD) with its basic statistical parameters (Log-Mean, Log-Median, Log-SD) and are shown (Table 1).

### 3.3. The Tail Index Estimation (TIE)

The results of the TIE program are shown in Table 2. For each assembly, the TIE program computed the three estimators—the Adjusted Hill (H) Estimator, the Moments (M) Estimator and the Kernel (K) Estimator—as detailed in Material and Methods. The conclusion of the analysis is given by the combined results of the three estimators given four possibilities: ‘Not Power Law’ (NLP), ‘Hardly Power Law’ (HLP), ‘Power Law’ (PL) and ‘Power Law with Divergent Second Moment’ (PL-DSM).

### 3.4. Positive and Negative Controls

The empirical coverage distributions of the two mapDamage control assemblies were computed: (1) human mitochondrial DNA (mtDNA) mapped with aDNA reads and (2) modern adenovirus reference mapped with present-day reads (MADV), for the purpose of comparison with the aDNA viral assemblies (Figure 5). Assuming that the Lander–Waterman model is a good approximation for these sequencing projects, it is expected that the empirical coverage distribution follows a negative binomial distribution. We tested this hypothesis using a two sample Kolmogorov–Smirnov test comparing the empirical coverage distribution with a simulated/bootstrapped negative binomial distribution with the same sample size as the empirical coverage distribution.

The results show that for the MADV, a negative binomial distribution with parameters (*r*,*μ*) = (117;21,125) and KS *p*-value 0.126 was obtained. For the mtDNA, a negative binomial distribution with parameters (*r*,*μ*) = (19;1410) and KS *p*-value 0.232 was obtained. In both cases, the test did not reject the null hypothesis with significance level *α* = 0.01 (99%).

### 3.5. Log-Normal Fitting Using the PLFit Program

We considered the cases that were classified as ’Not Power Law’ (NPL) and ‘Hardly Power Law’ (HPL) and tested if, in some of these cases, the empirical coverage distribution could be well approximated by a (discretized) log-normal distribution. We perform the comparison by first applying Vuong’s test to a power law fitting versus a log-normal fitting to the same empirical distribution. Then, we apply the Bootstrap test to both fittings to obtain two estimates for the goodness-of-fit (GOF). The comparison between the obtained *p*-values complementing the information of the estimators was computed by the TIE program. The results are summarized in Table 3. First, we note that the *p*-values of the Bootstrap test are relatively high, in accordance with the TIE results. Second, due to the simulations needed to compute these *p*-values, there is some degree of uncertainty of about 0.01 in all cases. Therefore, to ensure a safe margin of error, we set the significance level to *α* = 0.1 (90%). The conclusion of these tests indicates that the random assemblies are well approximated by log-normal distributions, with very good agreement in the cases of HPV random and HSV random.

## 4. Discussion

The basic assumption of genome assembly by mapping to a reference sequence is that it follows a Poisson distribution [13]. In a Poisson distribution, the mean and variance are assumed to be equal. When the variance exceeds the mean, this indicates overdispersion. Overdispersion refers to a situation where the observed variance of a data set is greater than what is expected under a particular statistical model. In general, overdispersion can be caused by several factors, such as the following: unobserved heterogeneity, clustering or correlation between observations, misspecification of the distribution, measurement errors and external covariates [26,27]. Overdispersion in genome assemblies occurs when the variability in the number of reads mapped to each genomic position exceeds what would be expected under a simple Poisson model, which assumes that the mean and variance of the read coverage are equal; however, in many biological contexts, the variance often exceeds the mean. In genome assemblies, overdispersion can arise from several sources, including the following: (1) uneven sequencing coverage, (2) repetitive sequences, (3) spurious mapping, (4) PCR amplification bias, (5) sequencing errors, (6) randomness in sampling reads, (7) variation in gene copy numbers, (8) fragmentation bias and (9) reference genome inaccuracies [28,29].

In the current study, we show that aDNA assemblies are overdispersed compared to modern DNA assemblies via reference mapping (Figure 2 and Figure 5). The size of the reads is to be taken into consideration in the case of aDNA because a very stringent filter for size might discard precious information, as is the case with smaller reads associated with smaller pathogen genomes, in particular, viral genomes [1,2]. The comparison of assemblies with real reference sequences versus random references shows that although both follow heavy-tailed distributions, random assemblies are well approximated by log-normal distributions, with very good agreement in the cases of HPV random and HSV random (Table 3). This might explain, at least in part, real reference assemblies, even with smaller read sizes, provided that the assemblies passed the Welch’s t-test, as shown by Ferreira et al. [2]. This also suggests that random assemblies are even more overdispersed than real assemblies and the removal of reads might not be necessary to obtain statistically significant results, in other words, assemblies where the signa-to-noise ratio is acceptable.

Our analysis shows that the coverage distributions of the real ancient assemblies (ADV, HSV and HPV) do not follow power laws nor log-normal laws and that the coverage distributions of the random controls are well approximated by log-normal laws (Table 2 and Table 3). On the other hand, the coverage distributions of the negative control parvovirus B19 (real and random) follow a power law with infinite variance (Figure 3g,h) while the coverage distributions of the mapDamage negative control with non-ancient DNA (modern ADV) and the mapDamage positive control (human mtDNA) (Figure 5) are well approximated by the negative binomial distribution, which is consistent with the predictions of the Lander–Waterman model [13].

Our present work addresses the problem of overdispersion in aDNA assemblies, particularly in what concerns the tails of distributions. This analysis might contribute to future research by providing statistical methods to help in research leading to the identification of viral remnants in aDNA samples. Overdispersion has a significant impact on the tails of statistical distributions, particularly in the context of genomic data analysis, where the distribution of read counts often exhibits higher variability than expected under simpler models like the Poisson distribution. In general, overdispersion can lead to heavier tails in the statistical distribution, meaning that extreme events (very high or very low values) occur more frequently than predicted by distributions without overdispersion. This has important consequences in a variety of fields, including genomics, epidemiology, and ecology, where understanding the behavior of the tails of distributions is crucial for modeling rare events or extreme observations. In general, overdispersion impacts the tails of statistical distributions leading to (1) heavier tails, (2) higher variability and extreme values, (3) challenges in modeling, (4) robustness of tail predictions and especially (5) power law behavior. Overdispersion can give rise to power law behavior in the tails of the distribution, where extreme values follow a power law decay rather than exponential decay. This is particularly relevant in contexts like biological networks or genomic data, where certain highly expressed genes or abundant sequences may appear disproportionately often [30].

It can be argued that the distribution of real data deviates from the ideal model (the Lander–Waterman model) because it is well known that aDNA data often have poor quality; however, it would be imprudent to discard such data solely because they fail to meet the ideal distribution criteria. Here, we demonstrate that, at least in the context of ancient virus data, a bona fide assembly deviates from the ideal model but at the same time is different from a noisy, full random model that fits a power law. We propose that an assembly with heavy tails that fit a power law should be considered too noisy and therefore cannot be statistically differentiated from a full random model. One solution, in this case, to avoid discarding precious aDNA data, is to filter out the read subsets (using different criteria such as size and quality) to reduce the heavy tails. In this way, even if a particular bona fide aDNA assembly does not perfectly fit the Lander–Waterman model, it still might not fit completely to a random model and therefore can be accepted.

A possible primary limitation of the study would be the relatively small sample size of the ancient assemblies. However, it must be emphasized that the small sample size of the aDNA assemblies does not affect the power law estimation using the TIE method of Voitalov et al. [22], in the sense that a larger sample size would not make a difference. As stated above, it is important to stress that based on any given finite sample, there is absolutely no way to tell how likely the hypothesis is that it was sampled from a regularly varying distribution. In other words, it is impossible to construct a hypothesis test (based on finite samples and *p*-values) to tell what distribution class (regularly varying or not) the sample is coming from. This is one of the main points of [22]. The procedure proposed in Voitalov et al. [22] is not a hypothesis test. The issue mentioned above is not a limitation of the method itself, but a mathematical limitation of the problem. There are some limitations in using the PLFit method for the comparison between power law fitting versus a log-normal fitting (see [22] Appendix D.3 for a thorough discussion about the PLFit method).

## 5. Conclusions

In summary, the analysis described above provides a classification of the empirical coverage distributions: (1) the coverage distributions of the real aDNA assemblies (ADV, HSV and HPV) do not follow power laws nor log-normal laws, (2) the coverage distributions of the random controls of aDNA assemblies are well approximated by log-normal laws, (3) the coverage distributions of the negative control parvovirus B19 (real and random) follow a power law with infinite variance and (4) the coverage distributions of the mapDamage negative control with non-ancient DNA (Modern ADV) and the mapDamage positive control (human mtDNA) are well approximated by the negative binomial distribution. We conclude that the tails of distributions of reads in a genome assembly by reference mapping can reveal the level of random effects and assess the quality of the assemblies. In addition to non-parametric tests for signal-to-noise ratio, the statistical distributions, as studied here, can contribute to the mitigation of problems related to spurious alignments in aDNA reconstructions and inference.

## Figures and Tables

**Figure 1 viruses-17-00195-f001:**
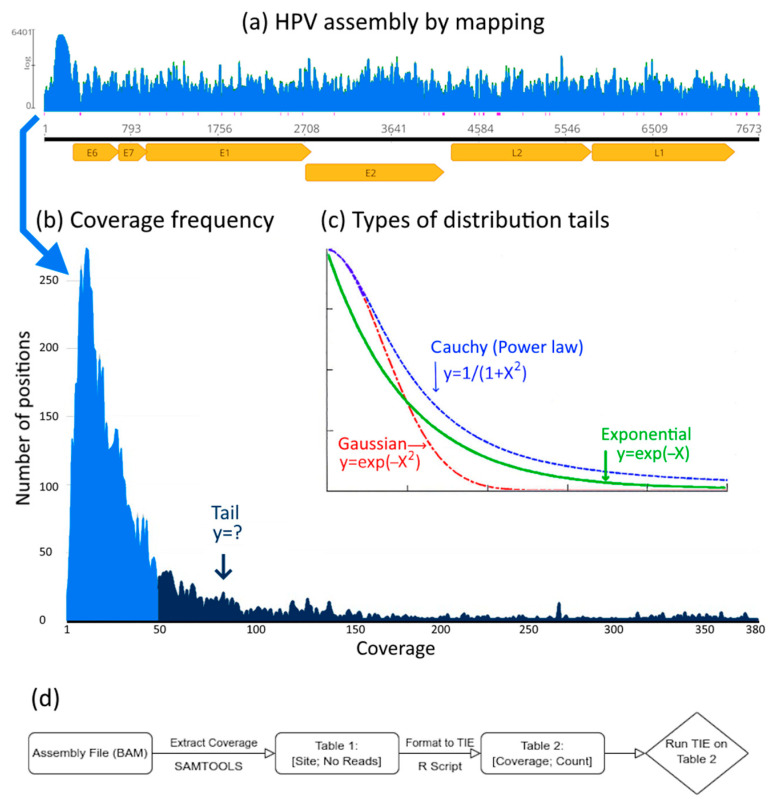
Distribution tails of genome assemblies. In (**a**) the coverage (number of reads per position, indicated by the blue area) along the HPV genome (with annotated genes in yellow) [2]. In (**b**) the coverage distribution (number of positions with given coverage, blue area and tail in dark blue). In (**c**) the different types of distribution tails show the difference between Gaussian (light-tailed, red line), exponential (light-tailed, green line) and a power law (Cauchy, blue line) a heavy-tailed distribution. Heavier tails indicate more random effects. In (**d**) the TIE workflow. Starting with the assembly (typically a BAM file), the coverage count is extracted using the SAMTOOLS program version 1.21. The output (Table 1) can be a TXT or CSV file with at least two columns: Site and Number of Reads, which covers Site. From Table 1, the infile for the TIE program is generated (Table 2). This can be a TXT or CSV file with exactly two columns (and no header): Coverage (integer values starting at 0) and Count (the number of sites with the given Coverage). The output file contains the Empirical Coverage Distribution. The TIE program runs using Table 2 as infile. The output is a PDF file with a plot of six graphs showing the results of the computations.

**Figure 2 viruses-17-00195-f002:**
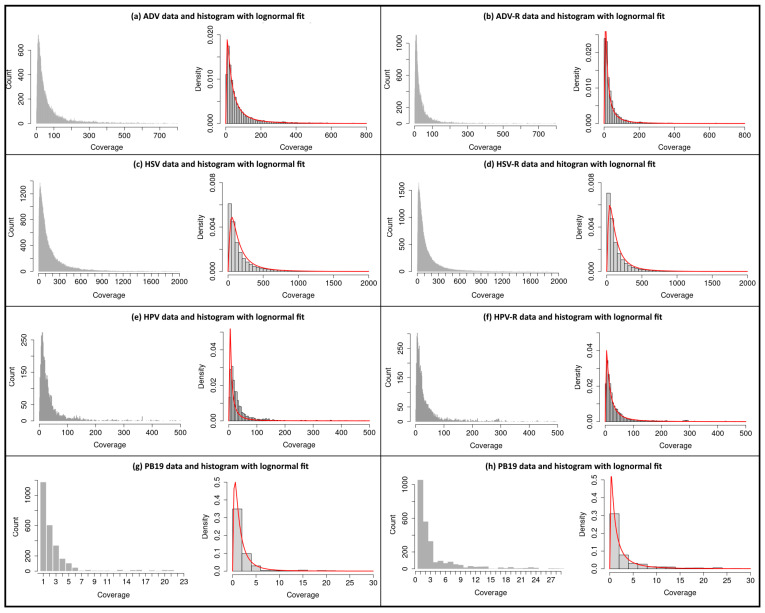
Coverage distributions of assemblies (BAM files) with corresponding log-normal fit. In (**a**) ADV and (**b**) the corresponding random “mock” control ADV-R. In (**c**) HSV and (**d**) the corresponding random “mock” control HSV-R. In (**e**) HPV and (**f**) the corresponding random “mock” control HPV-R. In (**g**) the negative control PB19 and (**h**) the corresponding random “mock” control PB19-R. The red lines indicate the theoretical lognormal curves that fit the data.

**Figure 3 viruses-17-00195-f003:**
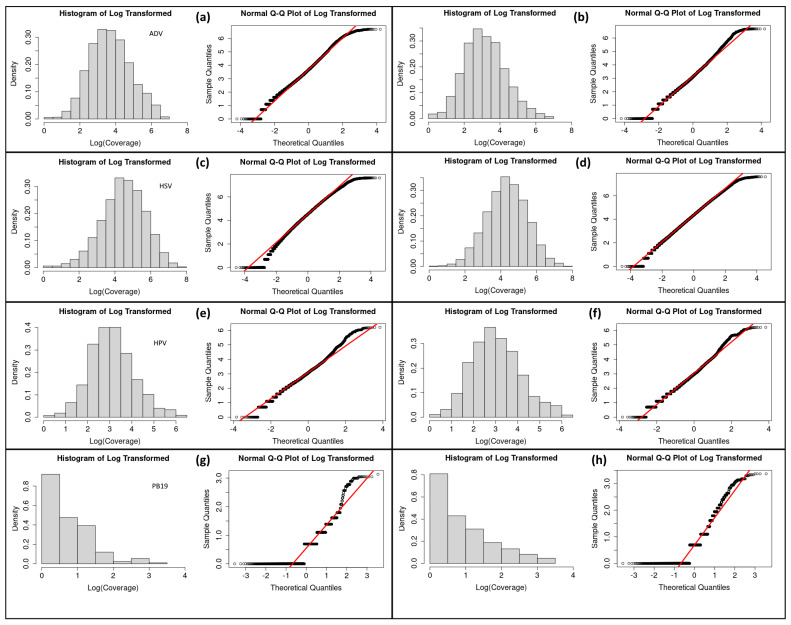
Histograms and Q-Q plots of log-transformed coverage distributions of ADV (**a**) and ADV random control (**b**), HSV (**c**) and HSV random control (**d**), HPV (**e**) and HPV random control (**f**) and negative control PB19 (**g**) and PB19 random control (**h**). The red lines indicate the expected results if the empirical distributions were Gaussian. The circles indicate the observed data.

**Figure 4 viruses-17-00195-f004:**
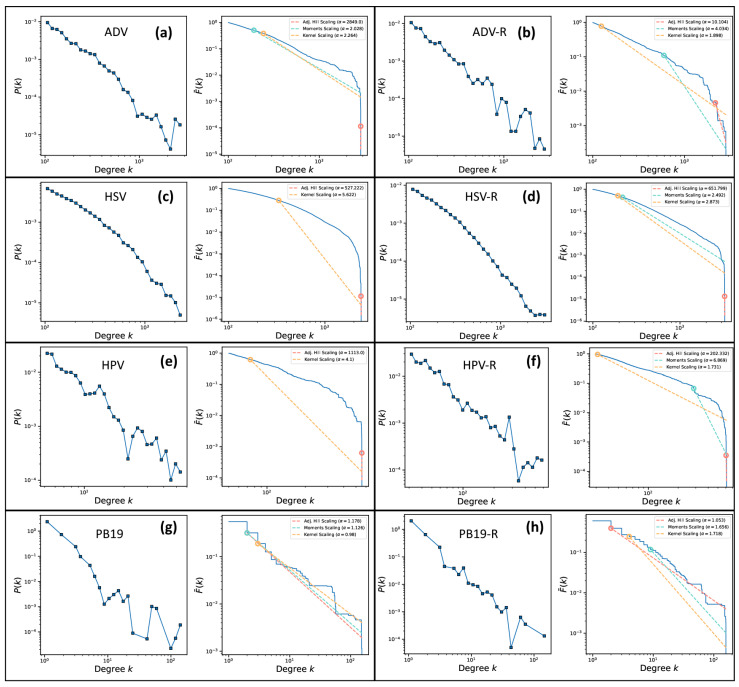
Log-log plots of the distribution (log *k*, log *P*(*k*)) and corresponding log-log plot of the complementary cumulative distribution function (blue lines) (log *k*, log *Ḟ*(*k*)) of ADV (**a**), ADV random (**b**), HSV (**c**), HSV random (**d**), HPV (**e**), HPV random (**f**), PB19 (**g**) and PB19 random (**h**), respectively.

**Figure 5 viruses-17-00195-f005:**
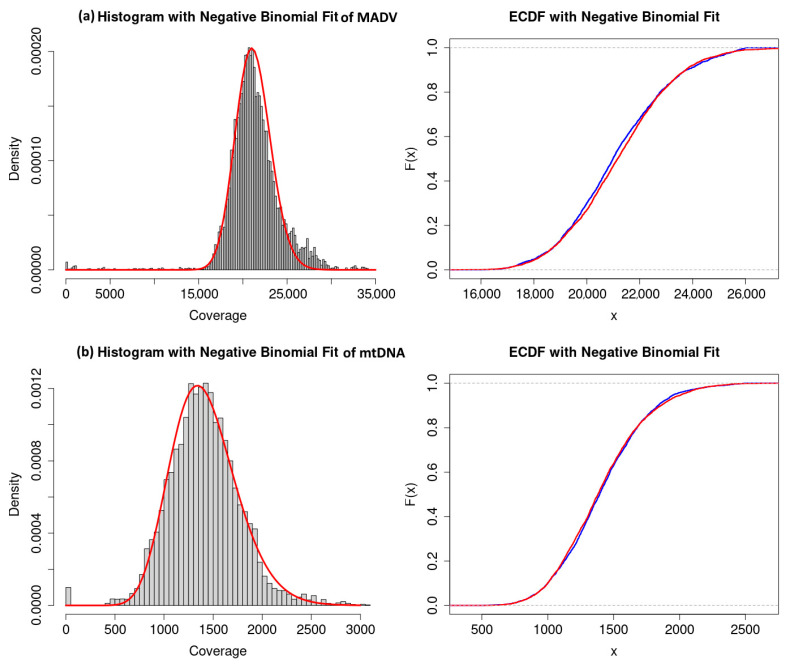
Histogram of coverage distributions and the empirical cumulative distribution function (ECDF) with negative binomial fit of modern ADV (MADV) (**a**) and mitochondrial DNA reference assembled with aDNA reads (**b**) used in [2] as mapDamage controls. On the left panels the red lines are the theoretical curves of the negative binomial distributions that fit the data. On the right panels, the blue lines are the empirical cumulative distributions, and the red lines are the negative binomial theoretical distributions that fit the data.

**Table 1 viruses-17-00195-t001:** Basic statistical parameters of the empirical coverage distributions. Columns 2 to 5 contain the mean, median, standard deviation (SD) and the number of reads (N) of the empirical coverage distributions, respectively. Columns 6 to 8 contain the mean, median and standard deviation of the log-transformed distributions (Log-Mean, Log-Median, Log-SD), respectively.

Assembly	Mean	Median	SD	N (reads)	Log-Mean	Log-Median	Log-SD
ADV	102.2	40	249.4	180,419	3.7	3.6	1.1
ADV-R	62.1	22	180.2	126,613	3.2	3.1	1.2
HSV	171.2	92	274.3	1,224,713	4.4	4.5	1.2
HSV-R	154.7	77	704.2	1,166,326	4.3	4.3	1.1
HPV	115.3	22	609.1	23,998	3.2	3.1	1.2
HPV-R	50.9	20	115.9	22,682	3.1	3.0	1.1
PB19	2.0	0	9.2	714	0.7	0.7	0.9
PB19-R	2.7	0	10.0	975	0.9	0.7	1.0

**Table 2 viruses-17-00195-t002:** Results of the TIE program. The first three columns contain the values of the three estimators, the Adjusted Hill Estimator (H), the Moments Estimator (M) and the Kernel Estimator (K). The last column contains the results of the analysis: ‘Not Power Law’ (NLP), ‘Hardly Power Law’ (HLP) and ‘Power Law with Divergent Second Moment’ (PL-DSM); see Section 2.1.2.

Assembly	Hill (H)	Moments (M)	Kernel (K)	Conclusion
ADV	0.000	0.493	0.449	NPL
ADV-R	0.099	0.248	0.517	HPL
HPV	0.001	−0.531	0.236	NPL
HPV-R	0.005	0.146	0.576	HPL
HSV	0.002	−0.431	0.172	NPL
HSV-R	0.002	0.401	0.348	HPL
PB19	0.849	0.888	1.034	PL-DSM
PB19-R	0.949	0.604	0.613	PL-DSM

**Table 3 viruses-17-00195-t003:** PLFit results of the comparison between power law fitting versus a log-normal fitting. The first column shows the *p*-values of Vuong’s test. A small *p*-value indicates that one of the distributions is closer to the true distribution. Columns 2 and 3 show the *p*-values of the Bootstrap test. Column 2 shows the *p*-values for the power law (PL) GOF and column 3 shows the *p*-values for the log-normal (LN) GOF. Significance level *α* = 0.1 (* one or both *p*-values are very close to the threshold α, giving a marginal rejection/non-rejection).

Assembly	Vuong’s	Bootstrap PL	Bootstrap LN	Conclusion
ADV	0.0	0.25	0.04	Not Reject PL (PL > LN)
ADV-R	1.7 × 10^−6^	0.09	0.15	* Not Reject LN (LN > PL)
HPV	1.4 × 10^−7^	0.04	0.12	* Reject both (LN > PL)
HPV-R	1.7 × 10^−8^	0.16	0.37	* Not Reject LN (LN > PL)
HSV	7.6 × 10^−4^	0.00	0.10	Reject both (LN > PL)
HSV-R	1.8 × 10^−9^	0.00	0.21	Not Reject LN (LN > PL)

## Data Availability

All data produced by analyses in this study are available upon request after acceptance of the manuscript for publication.

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
