# Peer review of "Statistical Distributions of Genome Assemblies Reveal Random Effects in Ancient Viral DNA Reconstructions"

_viruses, 2025, doi:10.3390/v17020195_

Round 1
Reviewer 1 Report
Comments and Suggestions for Authors
In this study, the authors have chosen a very specific and narrow research focus. However, their analyses are thorough, meticulous, and rigorous. In fact, this paper aligns closely with my own research background. The introduction, structure, and argumentation are all well-written, and I personally appreciate this style of research. While the results may lack elaborate visualizations, they often provide a more reliable reference for future researchers. Thus, I am inclined to support the publication of this paper in principle. However, there are certain deficiencies in its current state. I recommend addressing the following aspects to make the study more comprehensive:
Major Comments
1. What significant consequences might arise when the distribution of real data deviates from the ideal model? It is well known that ancient DNA (aDNA) data often suffer from poor quality, but it would be imprudent to discard such data solely because they fail to meet the ideal distribution criteria.
2. Could the authors establish a more detailed and standardized workflow to address similar issues? Such a workflow would help standardize the analytical process, providing clear guidelines on when data distributions are still usable and when they are entirely unsuitable for analysis.
Minor Comment
1. In the sentence:
"The size of the reads is to be taken into consideration in the case o aDNA because a very stringent filter for size might discard precious information,"
is "case o aDNA" a typographical error?
2. Line index should be added across the manuscript.
Author Response
Comment 1: In this study, the authors have chosen a very specific and narrow research focus. However, their analyses are thorough, meticulous, and rigorous. In fact, this paper aligns closely with my own research background. The introduction, structure, and argumentation are all well-written, and I personally appreciate this style of research. While the results may lack elaborate visualizations, they often provide a more reliable reference for future researchers. Thus, I am inclined to support the publication of this paper in principle. However, there are certain deficiencies in its current state. I recommend addressing the following aspects to make the study more comprehensive:
Response 1: We thank the reviewer for time and care in the analysis of our manuscript. The suggestions are welcome. We wish to congratulate the reviewer for letting her/his name appear in the open review system. This level of transparency is of great benefit to the scientific community and Viruses is to be congratulated as well for supporting this.
Major Comments
Comment 2. What significant consequences might arise when the distribution of real data deviates from the ideal model? It is well known that ancient DNA (aDNA) data often suffer from poor quality, but it would be imprudent to discard such data solely because they fail to meet the ideal distribution criteria.
Response 2: As a matter of fact, the idea that, as the reviewer points out “it would be imprudent to discard such data solely because they fail to meet the ideal distribution criteria” is the main message of this work. We demonstrate that, at least in the context of ancient virus data, a bona fide assembly deviates from the ideal model (the Lander-Waterman model) but at the same time is different from a noisy, random model that fits a Power law. We propose however that an assembly with heavy tails that fit a Power law should be considered too noisy and therefore cannot be statistically differentiated from a full random model. One solution in this case, to avoid discarding precious aDNA data, is to filter out the read subsets (using different criteria such as size and quality) in order to reduce the heavy tails. In this way even if a particular bona fide aDNA assembly does not fit perfectly the Lander-Waterman model it still might not fit completely to a random model and therefore can be accepted. We included this comment in the revised version.
Comment 3. Could the authors establish a more detailed and standardized workflow to address similar issues? Such a workflow would help standardize the analytical process, providing clear guidelines on when data distributions are still usable and when they are entirely unsuitable for analysis.
Response 3. A figure with the workflow has been added in the revised version that addresses this comment.
Minor Comment
Comment 4. In the sentence: "The size of the reads is to be taken into consideration in the case o aDNA because a very stringent filter for size might discard precious information," is "case o aDNA" a typographical error?
Response 4. This is a typographic error and was corrected in the revised version.
Comment 5. Line index should be added across the manuscript.
Response 5. Line numbers were added in the revised version.
Reviewer 2 Report
Comments and Suggestions for Authors
The manuscript by Fernando Antoneli and colleagues presents a comparative analysis of ancient and modern human viral sequence assemblies. The authors aim to elucidate patterns in the statistical distribution of mapped reads to predict the reliability of these assemblies. Specifically, they analyze four DNA assemblies containing sequences of ancient viruses, along with their corresponding reference assemblies, and investigate overdispersion, particularly in the tails of the distributions. Given the rapid advancements in paleovirology, the problems addressed in this study are of significant importance and the conducted analysis is highly relevant.
The manuscript is logically structured, follows a clear and accessible format, and employs appropriate figures. The methods used are rigorous and suitable for the research questions, and the results presented are convincing.
The primary limitation of the study is the relatively small sample size of ancient assemblies; however, this is understandable given the current scarcity of available data. It is crucial that the authors acknowledge and thoroughly discuss this and other limitations within the text. Additionally, the introduction would benefit from a more detailed discussion of the problem’s significance in the context of paleovirology and recent relevant advances in the field.
With these revisions, the study should be considered for publication in the journal Viruses.
Author Response
Comment 1. The manuscript by Fernando Antoneli and colleagues presents a comparative analysis of ancient and modern human viral sequence assemblies. The authors aim to elucidate patterns in the statistical distribution of mapped reads to predict the reliability of these assemblies. Specifically, they analyze four DNA assemblies containing sequences of ancient viruses, along with their corresponding reference assemblies, and investigate overdispersion, particularly in the tails of the distributions. Given the rapid advancements in paleovirology, the problems addressed in this study are of significant importance and the conducted analysis is highly relevant.
The manuscript is logically structured, follows a clear and accessible format, and employs appropriate figures. The methods used are rigorous and suitable for the research questions, and the results presented are convincing.
Response 1. We thank the reviewer for comments and suggestions.
Comment 2. The primary limitation of the study is the relatively small sample size of ancient assemblies; however, this is understandable given the current scarcity of available data. It is crucial that the authors acknowledge and thoroughly discuss this and other limitations within the text.
Response 2. The small sample size of the ancient assemblies does not affect the power law estimation using the TIE method of Voitalov et al. [Ref. 22 in the revised version: Voitalov, I.; van der Hoorn, P.; van der Hofstad, R.; Krioukov, D. Scale-Free Networks Well Done. Phys. Rev. Res. 2019, 1, 033034, doi:10.1103/PhysRevResearch.1.033034.], in the sense that a larger sample size would not make a difference. This is mentioned in the main text: “It is important to stress that based on any given finite sample, there is absolutely no way to tell how likely the hypothesis is that it was sampled from a regularly varying distribution.” In other words, it is impossible to construct a hypothesis test (based on finite samples and p-values) to tell what distribution class (regularly varying or not) the sample is coming from. This is one of the main points of [Ref. 22]. See the reference for a technical explanation. Notice that the procedure proposed in [Ref. 22] is NOT a hypothesis test.
As for possible limitations of the study, the issue just mentioned above is a limitation not of the method itself, but a mathematical limitation of the problem. There are some limitations in using the PLFit method for the comparison between Power Law fitting versus a Log-Normal fitting [see Ref. 22] Appendix D.3 for a thorough discussion about the PLFit method).
Comment 3. Additionally, the introduction would benefit from a more detailed discussion of the problem’s significance in the context of paleovirology and recent relevant advances in the field.
Response 3. Additional work has been cited with more detail in the introduction.
Comment 4. With these revisions, the study should be considered for publication in the journal Viruses.
Response 4. Once again, we thank the reviewer.
Round 2
Reviewer 1 Report
Comments and Suggestions for Authors
Thanks for your effort in revising it. I do not have further comments on the manuscript.